# Dual Effects of Korean Red Ginseng on Astrocytes and Neural Stem Cells in Traumatic Brain Injury: The HO-1–Tom20 Axis as a Putative Target for Mitochondrial Function

**DOI:** 10.3390/cells11050892

**Published:** 2022-03-04

**Authors:** Minsu Kim, Sunhong Moon, Hui Su Jeon, Sueun Kim, Seong-Ho Koh, Mi-Sook Chang, Young-Myeong Kim, Yoon Kyung Choi

**Affiliations:** 1Bio/Molecular Informatics Center, Department of Bioscience and Biotechnology, Konkuk University, Seoul 05029, Korea; kjmmmmm6@konkuk.ac.kr (M.K.); sunhong95047@konkuk.ac.kr (S.M.); heeth313@konkuk.ac.kr (H.S.J.); rlatndms0325@konkuk.ac (S.K.); 2Department of Neurology, Hanyang University Guri Hospital, Guri 11923, Korea; ksh213@hanyang.ac.kr; 3Department of Oral Anatomy, Seoul National University School of Dentistry, Seoul 03080, Korea; mschang@snu.ac.kr; 4Department of Molecular and Cellular Biochemistry, School of Medicine, Kangwon National University, Chuncheon 24341, Korea; ymkim@kangwon.ac.kr

**Keywords:** astrocyte, Korean red ginseng extract, heme oxygenase-1, traumatic brain injury, animal study, neurology

## Abstract

Astrocytes display regenerative potential in pathophysiologic conditions. In our previous study, heme oxygenase-1 (HO-1) promoted astrocytic mitochondrial functions in mice via the peroxisome-proliferator-activating receptor-γ coactivator-1α (PGC-1α) pathway on administering Korean red ginseng extract (KRGE) after traumatic brain injury (TBI). In this study, KRGE promoted astrocytic mitochondrial functions, assessed with oxygen consumption and adenosine triphosphate (ATP) production, which could be regulated by the translocase of the outer membrane of mitochondria 20 (Tom20) pathway with a PGC-1α-independent pathway. The HO-1–Tom20 axis induced an increase in mitochondrial functions, detected with cytochrome c oxidase subunit 2 and cytochrome c. HO-1 crosstalk with nicotinamide phosphoribosyltransferase was concomitant with the upregulated nicotinamide adenine dinucleotide (NAD)/NADH ratio, thereby upregulating NAD-dependent class I sirtuins. In adult neural stem cells (NSCs), KRGE-treated, astrocyte-conditioned media increased oxygen consumption and Tom20 levels through astrocyte-derived HO-1. HO inactivation by Sn(IV) protoporphyrin IX dichloride in TBI mice administered KRGE decreased neuronal markers, together with Tom20. Thus, astrocytic HO-1 induced astrocytic mitochondrial functions. HO-1-related, astrocyte-derived factors may also induce neuronal differentiation and mitochondrial functions of adult NSCs after TBI. KRGE-mediated astrocytic HO-1 induction may have a key role in repairing neurovascular function post-TBI in peri-injured regions by boosting astrocytic and NSC mitochondrial functions.

## 1. Introduction

In central nervous system (CNS) injuries, Korean red ginseng extract (KRGE) and its components (e.g., ginsenoside) have favorable effects on neurovascular regeneration and anti-inflammation [1]. Several ginsenosides have improved behavior in animal models of neurological deficits [1]. Ginsenoside Rg1 is involved in neurotrophic factor-mediated adult hippocampal neurogenesis and exhibits antidepressant activity [2]. Ginsenoside Rb1 may be protective against traumatic brain injury (TBI) by enhancing the gap junction [3]. KRGE pretreatment protects against acute sensorimotor deficits and promotes its long-term recovery after ischemic stroke through the nuclear factor erythroid 2-related factor 2 (Nrf2) pathway [4].

Neuroinflammatory brain injury, including TBI, induces the altered remodeling of astrocyte mitochondrial networks [5,6]. Heme oxygenase-1 (HO-1), a downstream target of Nrf2, is a heme-degrading enzyme that produces carbon monoxide (CO), iron, and biliverdin. In addition, biliverdin can be converted into bilirubin by biliverdin reductase [7]. HO-1 has cytoprotective effects on mitochondrial oxidative stress induced by mucosal injury in rats [8].

Nicotinamide adenine dinucleotide (NAD) salvage synthesis in mammals warrants the enzymatic activity of intracellular nicotinamide phosphoribosyl transferase (Nampt) [9]. The deletion of intracellular Nampt in the projection neurons of adult mice impairs mitochondrial function and neuromuscular junction synaptic transmission [5]. Silent information regulators (i.e., sirtuins (SIRTs)) comprise seven types (i.e., SIRT 1–7): class I (e.g., SIRT 1–3), class II (e.g., SIRT4), class III (e.g., SIRT5), and class IV (e.g., SIRT6 and SIRT7), which are NAD-dependent proteins [10,11,12]. SIRT1 may trigger peroxisome proliferators-activated receptor γ-coactivator-1α (PGC-1α) activation through deacetylation in skeletal muscles and provide beneficial effects to mitochondrial biogenesis [13,14]. However, pyruvate induces mitochondria biogenesis in PGC-1α null mouse-derived myoblasts [15], which implies the existence of a molecular pathway involved in PGC-1α-independent mitochondria biogenesis.

Astrocytes in the CNS contribute to neuroprotection and energy-metabolic activity in CNS-related pathological states [5,16]. Our previous study [6] demonstrated that KRGE induces HO-1 production in astrocytes post-TBI, which exerts crucial roles in mitochondrial activity, partly through PGC-1α. However, the roles of KRGE-induced astrocytic HO-1 in the mitochondrial components of astrocytes and neural stem cells (NSCs) have not been well studied. In this study, our novel finding was that astrocytic mitochondrial functions are considerably associated with the KRGE-mediated upregulation of translocase of the outer membrane of mitochondria 20 (Tom20) through the Nrf2–HO-1 axis in which the pathway involves a PGC-1α-independent mechanism. Tom20 is transcribed in the nucleus and becomes targeted to the mitochondrial outer membrane and thereby involves the translocation of major protein precursors [17].

To investigate the unknown molecular link between HO-1 and Tom20, we evaluated NAD-dependent class I SIRTs (i.e., SIRT1, SIRT2, and SIRT3). We checked the signaling cascade among HO-1, Nampt, SIRT1, SIRT2, SIRT3, and Tom20 in KRGE-mediated mitochondrial functions in astrocytes during oxygen–glucose deprivation (OGD), followed by recovery (OGD/R). KRGE induced HO-1 expression and the consequent upregulation of Nampt-class I SIRTs–Tom20 in OGD/R-conditioned astrocytes. In an in vivo TBI model, HO inhibition by Sn(IV) protoporphyrin IX dichloride (SnPP) injection diminished the expressions of KRGE-mediated, mitochondria-related proteins such as Nampt, SIRT1, SIRT2, SIRT3, and Tom20. Moreover, KRGE-mediated HO-1 induction in astrocytes also triggered intercellular communication by enhancing the mitochondrial activation, neuronal differentiation, and proliferation of adult NSCs.

## 2. Materials and Methods

### 2.1. Materials

Nicotinamide (NAM) (Merch Millipore, Temecular, CA, USA) and SnPP (Frontier Scientific, Logan, UT, USA) were prepared. KRGE was obtained from the Korea Ginseng Cooperation (Daejeon, Korea). KRGE stock solution was made in filtered distilled water, aliquoted, and maintained at −25 °C with light protection.

### 2.2. Animals

We purchased C57BL/6 male mice from Joong Ah Bio, Inc. (Suwon, Korea) and maintained them under standard conditions with food and water available ad libitum. The experiments were approved by the Animal Ethics Committee of the Kangwon National University (Chuncheon, Korea; approval number KW-181119-2). This investigation conformed to the Guide for the Care and Use of Laboratory Animals published by the United States National Institutes of Health. KRGE (0.015 mg/mL) was administered to post-TBI mice through their drinking water for three days. The mice in control conditions were only administered water. The mice were intraperitoneally introduced with 50 μmol/kg or 75 μmol/kg SnPP (i.e., an HO inhibitor), followed by TBI.

### 2.3. Controlled Cortical Impact Model for TBI

Eight-week-old young male C57Bl/6 mice were anesthetized with 2% inhaled isoflurane in a 7:3 mixture of nitrous oxide (N_2_O) and oxygen (O_2_) utilizing an isoflurane vaporizer (VetEquip, Inc., Livermore, CA, USA). They were positioned in stereotaxic apparatus (RWD Life Science, Shenzhen, China). We accomplished craniotomy at the somatosensory cortex, approximately 5 mm over the right hemisphere, by using a drill. By using a controlled cortical impact device (Leica Biosystems, Buffalo, NY, USA), a 3 mm flat-tip impactor was accelerated to a depth of 2 mm at a velocity of 5 m/s. The brains were obtained from mice three days post-injury. The animals were randomly assigned to the TBI group. The sham groups only underwent craniotomy.

### 2.4. Brain Tissues Preparation and Immunohistochemistry

For the immunohistochemistry procedure, the mice were anesthetized utilizing isoflurane (1.5%) and N_2_O gas and transcardially perfused with saline. Utilizing a prime cutting temperature compound, the brain tissues in a −70 °C freezer (Thermo Fisher Scientific, Carthage, MO, USA) were sectioned into 20 μm samples using a cryostat (Thermo Fisher Scientific). To detect Nestin and glial fibrillary acidic protein (GFAP), these sections were incubated with 4% paraformaldehyde for 15 min and washed in this order for a 10 min interval: phosphate-buffered saline (PBS) −0.1% Tween20 (PBST) → 0.2% PBST → 0.1% PBST. Slides were incubated with 3% bovine serum albumin (BSA) for 1 h. The sections were subjected to antibodies such as mouse anti-Nestin (1:200; Abcam, Cambridge, UK) and rabbit anti-GFAP antibody (1:300; Abcam) in PBST (0.1% triton X-100 in PBS) at 4 °C for 16 h. To detect Tom20, brain slides were incubated with cold acetone for 15 min and dried for 5 min. We consequently washed them, consecutively with 0.1% → 0.2% → 0.1% (vol/vol) Tween 20 in PBS for 10 min each and incubated with 3% BSA for 1 h. The sections were incubated overnight with two antibodies, including rabbit anti-Tom20 (1:200, Abcam) with mouse anti-Nestin (1:200, Abcam), mouse anti-GFAP antibody (1:300, BD Biosciences), mouse anti- neuronal nuclear protein (NeuN) antibody (1:200, Millipore) or mouse anti-heme oxygenase 1 (anti-HO-1) antibody (1:100, Abcam) in PBST (0.1% triton X-100 in PBS) at 4 °C for overnight. After washing, they were incubated in a mixture of tetramethylrhodamine (TRITC)-conjugated donkey immunoglobulin G (IgG) (1:200; Jackson ImmunoResearch, West Grove, PA, USA) and fluorescein isothiocyanate (FITC)-conjugated donkey IgG (1:200; Jackson ImmunoResearch) as secondary antibodies for 1 h at room temperature. After washing with 0.1% → 0.2% → 0.1% (vol/vol) Tween 20 in PBS for 10 min, the sections were mounted with a solution (Fluoro-Gel II with 4′,6-diamidino-2-phenylindole (DAPI); Electron Microscopy Sciences, Hatfield, PA, USA). The visualized images were obtained from an inverted phase contrast microscope (Eclipse T*i*2-U; Nikon, Tokyo, Japan).

### 2.5. Cell Culture and Hypoxia

Primary human brain astrocytes were supplied from the Applied Cell Biology Research Institute (Kirkland, WA, USA). Cells were incubated in Dulbecco’s modified Eagle medium (DMEM) (HyClone, Omaha, NE, USA), added with 10% fetal bovine serum (FBS, Corning, NY, USA) and 1% penicillin streptomycin solution (HyClone). We cultured 43- to 55-day-old adult rat NSCs (Merck Sigma, Burlington, MA, USA) in proliferation media (StemCell, Vancouver, BC, Canada) on a laminin-coated dish (Sigma-Aldrich, Burlington, MA, USA). Upon reaching 70% density, these cells were subjected to hypoxia for 4 h. To simulate hypoxia, adult NSCs’ media were replaced with proliferation media, and cells were induced by perfusing 90% nitrogen (N_2_), 5% carbon dioxide (CO_2_), 5% hydrogen (H_2_)-containing gas for 15 min in a hypoxia chamber (Billups-Rothenberg, Del Mar, CA, USA), which was consequently occluded for 4 h at 37 °C in an incubator. The media were replaced with differentiation media (StemCell) and astrocyte-conditioned media (ACM) in a 1:1 ratio. These cells were incubated at 37° C in a normal oxygen (O_2_) incubator (Thermo Fisher Scientific) for 4 days.

### 2.6. Oxygen–Glucose Deprivation and Astrocytes Media Preparation

Upon reaching 80% density, we incubated primary human brain astrocytes with 0% FBS in no-glucose DMEM media (Thermo Fisher Scientific). OGD was induced by perfusing 90% N_2_, 5% CO_2_, 5% H_2_-containing gas for 15 min in a hypoxia chamber (Billups-Rothenberg) with consequent occlusion for 8 h. After removing the dishes from the chamber, these cells’ media were replaced into 0% FBS-containing DMEM (HyClone) with distilled water or 250 μg/mL KRGE for 24 h (OGD/R) at 37 °C in a normal O_2_ incubator (Thermo Fisher Scientific). The astrocytes’ media were collected and centrifuged at 1500 rpm for 5 min. The supernatant was kept in a −70 °C deep freezer for ACM.

### 2.7. Western Blot Experiments

Protein Extraction Agent (Elpis-Biotech, Daejeon, Korea) was utilized for whole-cell lysis. Proteins from the cell lysates were added with the sodium dodecyl sulfate (SDS) sample buffer (glycerol 10% (*v*/*v*), Tris-Cl pH 6.8, SDS 2% (*w*/*v*) β-mercaptoethanol 1% (*v*/*v*), and bromophenol blue) and exposed to 98 °C for 5 min. The mixed samples underwent SDS–polyacrylamide gel electrophoresis. Proteins transferred onto polyvinylidene difluoride membranes (Merch Millipore) were blocked in Tris-buffered saline containing 0.1% Tween 20 and 5% skim milk (BD Difco, Burlington, NC, USA). The membranes were incubated overnight with primary antibodies at 4 °C. We used the following primary antibodies: Tom20 (1:3000, Abcam); Nampt (1:3000, AdipoGen Life Sciences, San Diego, CA, USA); cytochrome c (1:3000, BD Biosciences); HO-1 (1:1000, BD Biosciences; 1:2000, Enzo Life Science, Farmingdale, NY, USA), SIRT1 (1:1000, Santa Cruz Biotechnology, Dallas, TX, USA), SIRT2 (1:3000, Abcam); SIRT3 (1:1000, Santa Cruz Biotechnology); nuclear factor erythroid 2-related factor 2 (Nrf2) (1:1000, Santa Cruz Biotechnology); PGC-1α (1:1000, Santa Cruz Biotechnology; 1:3000, Thermo Fisher Scientific); cytochrome c oxidase subunit 2 (MTCO 2) (1:1000, Santa Cruz Biotechnology); Nestin (1:2000, Abcam); NeuN (1:3000, Millipore), growth-associated protein 43 (GAP43) (1:1000, Santa Cruz Biotechnology); and β-actin (1:8000, Sigma-Aldrich, Saint Louis, MO, USA). After washing, the membranes were incubated with peroxidase-conjugated secondary antibodies (1:8000, Thermo Fisher Scientific). After washing, ECL (Elpis-Biotech, Daejeon, South Korea) was used as treatment on the membranes, and images were detected using detection equipment (Fusion Solo-Vilber Lourmat, Collegien, France).

### 2.8. Transfection in Astrocytes

Upon reaching 70% confluence, we transiently transfected the astrocytes with small interfering ribonucleic acids (siRNAs) for Nampt, SIRT1, SIRT2, SIRT3, Tom20, HO-1 (50 nM, Santa Cruz Biotechnology), or a negative control (50 nM, Thermo Fisher Scientific) by using RNAiMax (Thermo Fisher Scientific), based on the manufacturer’s instructions. After approximately 14 h of recovery, the cells were incubated in OGD for 8 h and treated with or without KRGE for 24 h in 0% FBS with 1% penicillin-streptomycin-containing DMEM.

### 2.9. Cell-Counting Kit-8 Assay for Proliferation and Viability

Upon reaching 80% confluency, human astrocytes in 12-well plates (Merck Sigma) were subjected to OGD for 8 h. The media were subsequently replaced with 0% FBS-containing DMEM (HyClone). Various concentrations of KRGE were added, and the cells were incubated for 23 h. Thereafter, 30 μL of a cell-counting kit-8 (CCK-8) agent (Dojindo, Fukuoka, Japan) was added to each well, and cells were incubated at 37 °C for 1 h. We used a plate reader (Epoch Microplate Spectrophotometer; BioTek, Santa Clara, CA, USA) to determine the absorbance at a wavelength of 450 nm. The background wave was determined at 630 nm and subtracted from the value measured at 450 nm.

### 2.10. Cytotoxicity Analysis by Cell Survival Ratio

A cytotoxicity assay was applied using a lactate dehydrogenase (LDH) assay kit (Merck Sigma). The astrocyte medium was subjected to centrifugation for 7 min at 7000 rpm and then shifted to 96-well plates (CM). The same volume (50 µL) of the medium from plates without cells was regarded as a blank CM. Astrocytes were washed with PBS and incubated with 500 µL of 5% triton X-100 (Sigma-Aldrich) in PBS at 37 °C for 20 min. The lysed cells were centrifuged for 5 min at 15,000 rpm. We transferred the supernatant (50 µL) to 96-well plates (WCL). The supernatant from plates without cells served as the blank WCL. A dye solution was mixed with the catalyst (45:1 ratio). The aforementioned mixed reagent was added to each of the assay wells on the top of the supernatant in fast sequence. The assay plates of total volume (100 µL) were incubated at room temperature with light protection for 20 min and read using a plate reader (Epoch Microplate Spectrophotometer; BioTek) at a wavelength of 490 nm. The values were introduced into the following equation: survival (%) = (WCL value–WCL blank)/([WCL value–WCL blank] + [CM value–CM blank]) × 100.

### 2.11. Intracellular Nicotinamide Adenine Dinucleotide/Nicotinamide Adenine Dinucleotide Hydrogen Assay

We quantified the intracellular NAD^+^ and NAD hydrogen (NADH) levels by using an NAD/NADH quantification kit (BioVision, Milpitas, CA, USA), based on the manufacturer’s instructions. Human astrocytes cultured in 60 mm dishes were extracted with 800 μL of the extraction buffer with two freeze/thaw cycles. To detect NADH, we heated 400 μL of the extracted samples to 60 °C for 30 min. Each 50 μL extracted sample with or without heating was added to a 96-well plate, followed by the addition of 50 μL of NAD^+^ cycling mix and subsequent incubation for 30 min. We added 5 μL of the NADH developer into the mix and incubated it at room temperature for 30 min. The plates were read at 450 nm (Epoch Microplate Spectrophotometer, BioTek). The NAD^+^/NADH ratio in the normoxia/recovery (i.e., control) group was set at “1,” and the values in other groups were adjusted to that of the control group.

### 2.12. Immunocytochemistry with Mitotracker Staining

Intracellular mitochondria activity was obtained from fluorescence imaging using 1 μM of Mitotracker (Thermo Fisher Scientific, Waltham, MA, USA), a mitochondrial membrane potential-sensitive dye. Astrocytes plated on 18 mm round coverslips in 12-well plates and were cultured to 80% confluency. The cells were subjected to distilled water or 500 μg/mL KRGE for 23.5 h, then 1 μM of Mitotracker was added for 30 min. Following washing with PBS, the cells were fixed in 4% paraformaldehyde for 10 min. They were washed with 0.1% → 0.2% → 0.1% (vol/vol) Tween 20 in PBS, each for 10 min. The blocking step, using 3% BSA prepared in PBST (i.e., PBS containing 0.1% Triton X-100), was introduced for 1 h at room temperature, followed by overnight incubation with rabbit anti-Tom20 antibody (1:3000, Abcam) prepared in PBST (PBS containing 0.1% Triton X-100) at 4 °C. After washing with 0.1% → 0.2% → 0.1% (vol/vol) Tween 20 in PBS, each for 10 min, the cells were incubated with FITC-conjugated donkey anti-rabbit IgG (1:300, Jackson ImmunoResearch) for 1 h at room temperature. We consequently placed a round cover glass over each well by using a mounting agent (Fluoro-Gel II with DAPI; Electron Microscopy Sciences). The fluorescent images were obtained using a microscope (Eclipse T*i*2-U; Nikon, Tokyo, Japan).

### 2.13. O_2_ Consumption

O_2_ consumption in live astrocytes or adult NSCs was detected by using the O_2_ Consumption Rate Assay Kit (Cayman, Ann Arbor, MI, USA). In astrocytes, cells with 70% confluent seeded on a 60 mm dish (Merch Sigma) were transfected with the negative control (si-NC) or Tom20 siRNA (si-Tom20) for 6 h. Cells were detached using trypsin and transferred into a 96-well black polystyrene microplate (Merch Sigma). The cells were subjected to OGD/R with 250 μg/mL KRGE or with distilled water for 24 h in serum-free DMEM media. An O_2_ sensor probe was added to each well. In adult NSCs, cells with 30–40% density in a laminin-coated, 96-well black polystyrene microplate (Merch Sigma) were subjected to 4 h of hypoxia, followed by 4 days of recovery. During recovery, the media were replaced with ACM and differentiation media (1:1 ratio), and NSCs were incubated for 4 days. Similar procedures in 96-well plates without a cell were conducted to evaluate the blank value. To detect the O_2_ consumption rate, an O_2_ sensor probe and mineral oil were added to the wells. Plates were introduced into the detector (Synergy H1, Hybrid Multi-Mode reader; BioTek) to obtain the absorbance value by using a filter combination and the emission and excitation wavelengths of 650 nm and 380 nm, respectively, at 37 °C for 75 min with 5 min interval. The value of absorbance in wells with cells and without cells at 15 min after O_2_ consumption rate detection in the hypoxia/recovery group (normoxia/recovery ACM-treated control) was set at “1,” and values in the other groups were adjusted to that of the control group.

### 2.14. Adenosine Triphosphate Levles

Intracellular adenosine triphosphate (ATP) levels were measured using an ATP colorimetric assay kit (BioVision). We transfected 70% of the confluent astrocytes with the Tom20 siRNA and exposed them to OGD/R with 250 μg/mL KRGE or with distilled water in serum-free DMEM media. They were lysed in the ATP assay buffer and centrifuged at 15,000 rpm for 5 min at 4 °C. The collected supernatant was combined with the same volume of the reaction mixture reagent (50 μL). The plates were incubated at room temperature for 30 min while being protected from light. We measured the absorbance at 570 nm by using a reader (Epoch Microplate Spectrophotometer; BioTek). The protein content in the lysed cells was quantified using bicinchoninic acid (Thermo Fisher Scientific) and measured at 562 nm using an Epoch Microplate Spectrophotometer (BioTek). The ATP levels/protein amount (ATP/protein amount) in the control group were subsequently set to 1, whereas the levels in the other groups were adjusted to that of the control group.

### 2.15. Data Analysis

The ImageJ (http://rsb.info.nih.gov/ij/ (accessed on 1 July 2021)) program was applied to detect the intensity of the protein band obtained from the Western blot experiments and immunofluorescence from immunocytochemistry. Values were analyzed using Prism 6 (GraphPad, San Diego, CA, USA). We conducted multiple comparisons using the one-way analysis of discrepancy and Tukey’s test (data are presented as the mean ± the standard deviation (SD)). Values of *p* < 0.05 were statistically significant (* *p* < 0.05, ** *p* < 0.01, and *** *p* < 0.001).

## 3. Results

### 3.1. KRGE-Induced Tom20 Is Coexpressed with Astrocytes in the Peri-Injured Regions of Traumatic Brain

KRGE was administered with drinking water for 3 days in 8-week-old mice with or without TBI (Figure 1a). At approximately bregma −1 to −2, the levels of protein (e.g., HO-1, Nrf2, Tom20, and PGC-1α) associated with mitochondrial functions were upregulated in peri-injured mouse brains subjected to KRGE administration after TBI (Figure 1a and Appendix A). We observed Tom20 protein expression in the KRGE-administered TBI peri-injured regions, which was colocalized with GFAP-stained astrocytes (Figure 1b,c), thereby suggesting KRGE in astrocytes may regulate Tom20 expression. KRGE-mediated HO-1 can be detected in GFAP-positive astrocytes after TBI [6]. Tom20 protein levels were detected in HO-1-positive regions in KRGE-administered TBI mouse brain (Appendix A). Therefore, HO-1 may be partially associated with Tom20 in astrocytes.

After TBI, damaged vascular cells may cause an O_2_- and glucose-deficient status [18]. Hence, we pretreated human astrocytes with OGD for 8 h, followed by recovery for 24 h (OGD/R) (Figure 1d). The treatment of human astrocytes with various concentrations of KRGE during recovery did not affect their cell viability, assessed with a cell-counting kit-8 (CCK-8) assay (Figure 1d). By contrast, KRGE increased the expressions of mitochondria-related proteins (e.g., Nrf2, HO-1, Tom20, and PGC-1α) in a concentration-dependent mode (Figure 1e). Nrf2 is an upstream factor for HO-1 [19]; therefore, we explored the impact of Nrf2 knockdown on protein levels of HO-1, Tom20, and PGC-1α by KRGE in OGD/R conditions (Figure 1f). Nrf2 knockdown effectively reduced the KRGE-mediated elevated expression of Nrf2, HO-1, Tom20, and PGC-1α using Nrf2-specific siRNA (si-Nrf2) in astrocytes (Figure 1f and Appendix A). KRGE-induced Tom20 could be partially associated with the Nrf2-HO-1 axis in the astrocytes located in peri-injured regions of TBI.

### 3.2. KRGE Induces HO-1-Mediated Tom20 in a PGC-1α-Independent Manner in Astrocytes

To determine the relationships among mitochondrial function-related proteins such as HO-1, Tom20, and PGC-1α, we transfected astrocytes with specific siRNA. Astrocytes subjected to KRGE in OGD/R elevated the expression of HO-1, Tom20, and PGC-1α. Moreover, these effects were effectively reduced by HO-1 knockdown using siRNA for HO-1 (si-HO-1) (Figure 2a). siRNA for PGC-1α (si-PGC-1α) did not reduce the expression of Tom20 and HO-1 (Figure 2b), whereas siRNA for the translocase of the outer membrane of mitochondria (si-Tom20) did not decrease the protein levels of PGC-1α and HO-1 (Figure 2c). Therefore, HO-1 regulated both Tom20 and PGC-1α based on different pathways. Furthermore, we examined the function of KRGE-mediated Tom20 by examining the components of mitochondrial electron transport chain (i.e., cytochrome c oxidase subunit 2 (MTCO2) and cytochrome c). Diminished Tom20 expression decreased MTCO2 and cytochrome c in OGD/R-conditioned astrocytes (Figure 2d). KRGE upregulated astrocytic mitochondrial functions partly through the HO-1-Tom20 pathway in a PGC-1α-independent manner.

### 3.3. KRGE Induces Mitochondrial Membrane Potential and ATP Production via Tom20

We examined the roles of Tom20 in the energy production of astrocytes exposed to KRGE in OGD/R conditions. The treatment of human astrocytes with si-Tom20 significantly reduced the intensity of KRGE-induced Tom20 in OGD/R conditions, assessed with immunocytochemistry (Figure 3a, upper). The density and shape of nuclei were nearly the same in all groups and showed a similar cell condition (Appendix A). si-Tom20 reduced the immunoreactivity of Mitotracker, a mitochondria membrane potential marker (Figure 3a, lower), which was critically increased by KRGE (Figure 3b,c). An interesting finding was that KRGE increased O_2_ consumption and ATP production via a Tom20-dependent pathway (Figure 3d,e). Thus, astrocyte-derived Tom20 may have a key role in mitochondrial activity and energy production in KRGE-treated OGD/R conditions.

### 3.4. KRGE Induces the HO-1-Nampt Circuit, Thereby Leading to an Elevated NAD^+^/NADH Ratio in Astrocytes

Astrocyte treatment with two combinatory HO metabolites—namely CO and bilirubin—increased Nampt levels [20]. Intracellular Nampt exerts an enzymatic activity that is responsible for salvaging the pathways of NAD^+^ synthesis, thereby activating NAD^+^-dependent protein deacetylases [10,20]. Thus, we determined the relationship between HO-1 and Nampt in the KRGE-treated OGD/R conditions. Astrocytes subjected to KRGE in OGD/R conditions revealed upregulated Nampt, which was significantly reduced by si-HO-1 (Figure 4a) or the HO inhibitor SnPP (Figure 4b). KRGE-induced HO-1 expression in OGD/R conditions was interestingly markedly blocked by si-Nampt (Figure 4c). Nampt crosstalk with HO-1 consequently leads to the upregulation of Tom20 protein (Figure 4b,c).

We subsequently examined the NAD^+^/NADH ratio after astrocyte transfection with or without si-Nampt in the KRGE-treated OGD/R conditions. KRGE promoted the intracellular NAD^+^/NADH ratio, which was blocked by si-Nampt (Figure 4d). To evaluate NAD^+^-dependent class I SIRTs, we used nicotinamide (NAM) as an inhibitor for class I SIRTs [21]. The treatment of 5 mM NAM with KRGE during recovery decreased the protein levels of class I SIRTs (i.e., SIRT1, SIRT2, and SIRT3) without altering the HO-1 expression (Figure 4e and Appendix A). The shorter size of the SIRT2 and SIRT3 bands may imply protein import into the mitochondria [11,22]. Similar to NAM, si-Nampt decreased the protein levels of SIRT1, concomitant with the shorter size of SIRT2 and SIRT3 (Figure 4f). Hence, KRGE may induce the HO-1-Nampt circuit and lead to the upregulation of SIRT1 levels and the cleavage form of SIRT2 and SIRT3.

### 3.5. KRGE Induces Tom20 Expression through the HO-1-Class I SIRTs Axis

HO-1 knockdown significantly decreased the KRGE-mediated SIRT1, SIRT2, and SIRT3 levels (Figure 5a). However, the knockdown of each class I SIRT (i.e., SIRT1, SIRT2, and SIRT3) did not reduce KRGE-induced HO-1 expression (Figure 5b), which suggests the role of the HO-1-class I SIRTs axis. The knockdown of class I SIRTs (i.e., si-SIRT1, si-SIRT2, or si-SIRT3) reduced the expression of Tom20. In addition, more potent Tom20 inhibition was observed after transfection with si-SIRT1 or si-SIRT3 than with si-SIRT2 (Figure 5b). Therefore, KRGE induced the HO-1–SIRTs–Tom20 axis. We conducted the experiments with TBI brains in vivo. HO inhibition by SnPP reduced the Nampt, SIRT1, SIRT2, SIRT3, and Tom20 levels, which were moderately increased with KRGE administration post-TBI (Figure 5c).

### 3.6. Astrocytic HO Enhances Crosstalk between Astrocytes and Adult NSCs

Energy-producing Tom20 was induced in the astrocytes of TBI plus KRGE mouse brains and was inhibited by SnPP-mediated HO inhibition, followed by TBI plus KRGE (Figure 6a). KRGE-administered TBI mouse brains revealed closely localized GFAP-positive astrocytes and Nestin-positive NSCs (Figure 6b). Thus, we examined the impact of energetic astrocytes by KRGE administration on neighboring cells such as NSCs. KRGE-treated ACM from the OGD/R conditions were transferred into adult NSCs in the hypoxia/recovery (H/R) phase for 4 days. The aforementioned ACM increased NeuN, the mature neuronal marker protein, in a concentration-dependent manner (Appendix A). However, the direct treatment of adult NSCs with various concentrations of KRGE did not induce NeuN expression in adult NSCs (Appendix A. We subsequently collected ACM from KRGE with or without SnPP during OGD/R conditions and transferred them into adult NSCs for 4 days. KRGE-treated ACM promoted the protein expression of mature neuronal markers such as NeuN and GAP43, compared to the non-KRGE-treated ACM (Figure 6c), which was blocked by ACM from the cotreatment of SnPP with KRGE in adult NSCs (Figure 6c). Of note, the direct treatment of SnPP with KRGE-treated ACM into NSCs did not significantly block the neuronal differentiation of NSCs (Figure 6c). Astrocyte-derived factors may collectively induce the mature neuronal differentiation of adult NSCs with KRGE treatment in OGD/R conditions, possibly through an HO-dependent pathway.

### 3.7. KRGE Induces NSCs’ Mitochondrial Functions through Astrocytic HO-1

We determined the mitochondrial functions in adult NSCs, which could be required for neuronal regeneration in peri-injured TBI brains. Tom20 expression was used for the mitochondrial component protein. We observed Tom20 induction in Nestin-positive NSCs upon administering KRGE in mice after TBI (Figure 7a); therefore, KRGE enhanced mitochondrial proteins such as Tom20 in NSCs and in astrocytes. The direct treatment of NSCs with KRGE did not induce Tom20 protein expression (Figure 7b). In contrast, treatment with KRGE-treated ACM significantly upregulated Tom20 expression (Figure 7c). Furthermore, we investigated whether transient HO-1 induction in astrocytes would elevate the neuronal differentiation and Tom20 expression in adult NSCs. The si-HO-1–KRGE-treated ACM significantly reduced the neuronal markers (i.e., Nestin, NeuN, and GAP43), as well as Tom20, compared to the si-control-KRGE-treated ACM on ACM transfer into adult NSCs for 4 days (Figure 7c). In addition, KRGE-treated ACM increased NSC proliferation, detected with the CKK-8 assay (Figure 7d) and O_2_ consumption (Figure 7e). The aforementioned effects were inhibited by the combination of si-HO-1-KRGE-treated ACM (Figure 7d,e). However, we could not identify any cell cytotoxicity by si-HO-1 ACM, assessed by the LDH assay (Figure 7f). The KRGE-mediated transfer of astrocyte-derived factors may enhance neuronal differentiation after TBI.

### 3.8. KRGE Induces Markers for Regeneration through HO Activation following TBI

The in vivo TBI model was injected with SnPP, followed by KRGE administration for 3 days. SnPP treatment diminished the expression of Tom20 and NeuN, particularly in the cornu ammonis 2 (CA2) region of the hippocampus, which increased after KRGE administration (Figure 8a). A recent report [23] reveals that mitochondrial pathways and respiration are overrepresented in CA2 cell bodies and dendrites. Tom20 surrounded the NeuN in the CA2 region of the KRGE intake mouse brains. The aforementioned effect was markedly blocked by 50 μmol/kg and 75 μmol/kg SnPP (Figure 8a). We observed greater reductions in Nestin, NeuN, and GAP43 by 75 μmol/kg SnPP than by 50 μmol/kg SnPP after subjecting KRGE-administered TBI brains (approximately bregma −1 to −2) to Western blotting (Figure 8b). Therefore, KRGE-induced HO activity can promote neuronal repair with elevated mitochondrial functions, followed by TBI.

## 4. Discussion

KRGE exerts beneficial effects on neuroinflammatory diseases in the CNS [1]. The transient induction of astrocytic HO-1 by KRGE after TBI may promote mitochondrial functions [6]. However, researchers have not yet established the molecular mechanisms by which KRGE induces astrocytic HO-1 in TBI. Li et al. [24] reported that quercetin induces Nrf2 nuclear translocation and enhances mitochondrial functions such as mitochondria membrane potential and ATP production in TBI. The Nrf2-HO-1 pathway exerts regenerative effects by modulating inflammation and vascular remodeling such as vasculogenesis and angiogenesis [7,19]. Our findings demonstrated that Nrf2 can act as an HO-1 regulator in TBI brains and astrocytes.

The Nrf2-HO-1 signaling cascade simultaneously activated PGC-1α and Tom20 on treating astrocytes with KRGE. The interaction between Nrf2 and PGC-1α signaling pathways is involved in mitochondrial biogenic activity [25]. PGC-1α is a master regulator of mitochondria biogenesis [13,26,27,28]. However, our results implied that KRGE induced the PGC-1α-independent and Tom20-dependent pathways, which act as other important regulation signals for O_2_ consumption and energy production in astrocytes and TBI.

KRGE induced the Nampt-HO-1 circuit in astrocytes. The combination of two HO metabolites—namely CO and bilirubin—upregulates Nampt expression [20]. In this study, we found that the upstream and circuit of HO-1 is Nampt. Therefore, intracellular NAD^+^ levels can affect mitochondrial functions through the positive circuit of Nampt-HO-1 and the consequent class I SIRTs activation.

To the best of our knowledge, this study is the first to identify the role of KRGE-induced Tom20 in astrocytic energy-producing steps by increasing the O_2_ consumption rate through HO-1-Nampt-class I SIRT (e.g., SIRT1, SIRT2, and SIRT3) pathways. NAM diminished the protein expression of KRGE-mediated SIRT1, SIRT2, and SIRT3; therefore, the inhibition of NAD^+^-dependent protein deacetylases reduced their expression as well. KRGE-induced Tom20 also regulates the mitochondrial membrane potential, thus establishing an active mitochondria-existing environment. We did not explore the mechanisms underlying the effect of SIRTs on Tom20 expression. We intend to determine the relationship between Nampt-mediated SIRTs activation and Tom20 in KRGE-treated astrocytes in a future study.

Astrocytes have emerged as active players in brain energy delivery by coordinately communicating with the neuronal system [29,30,31], thereby leading to neurovascular repair after CNS damage [16,30]. The transfer of ACM into NSCs regulates NSC proliferation and differentiation [32]. Researchers have not investigated the interaction between astrocytes and neuronal system in TBI through astrocytic HO-1. We hypothesized that secreting factor(s) originating from astrocytic HO-1 in the presence of KRGE may facilitate neuronal repair post-TBI. The inhibition of or reduction in astrocytic HO-1 affects the differentiation and proliferation of adult NSC. In addition, TBI brains with KRGE administration showed NeuN expression in the CA2 region, whereas this effect was blocked by SnPP cotreatment. Therefore, KRGE may accelerate astrocyte-NSCs communication, partly through astrocytic HO-1.

The HO-1/CO-Nampt-SIRT1 pathway increases vascular endothelial growth factor secretion from astrocytes [20]. This finding demonstrates facilitated signaling for angiogenesis, neurogenesis, and vascular homeostasis [33,34,35]. Taken together, as a putative therapeutic agent, KRGE may enhance neurovascular regeneration after TBI by mediating astrocytic mitochondria activity through HO-1-Tom20.

## 5. Conclusions

In this study, KRGE-treated astrocytes upregulated the proliferation and neuronal differentiation of adult NSCs, supposedly through astrocyte–neuronal system cooperation. Our results suggested that KRGE likely strengthens astrocyte mitochondrial function by HO-1 induction. In addition, HO-1-induced astrocytes contribute to the neuronal differentiation of adult NSCs. The HO-1–Tom20 axis may act on mitochondrial functions in intracellular and intercellular pathways. Taken together, the KRGE-induced HO-1 pathway may contribute to mitochondrial activity and energy production in astrocytes, and thereby lead to the potential improvement of neurovascular repair post-TBI.

## Figures and Tables

**Figure 1 cells-11-00892-f001:**
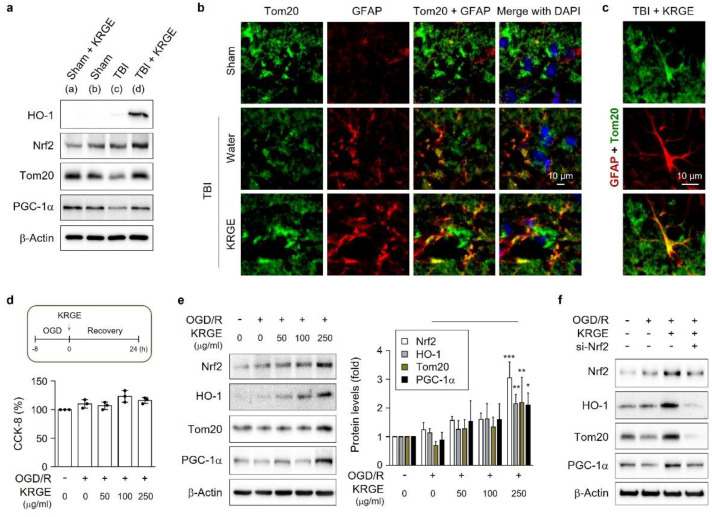
Korean red ginseng extract (KRGE)-induced translocase of the outer membrane of mitochondria 20 (Tom20) is co-expressed with astrocytes in peri-injured regions of a traumatic brain. (**a**) Protein expression from brain tissues (approximately bregma −1 to −2), assessed with Western blotting. β-Actin is the internal control (*n* = 4 per each group). (**b**,**c**) Representative images of the Tom20 (green) and glial fibrillary acidic protein (GFAP) (red) in a mouse brain obtained from the sham, traumatic brain injury (TBI), and TBI followed by KRGE (TBI + KRGE) treatment groups (*n* = 3 per group). 4′,6-Diamidino-2-phenylindole (DAPI, blue) is used for nucleus detection (the scale bar = 10 μm). (**d**) The schematic figure displays the conditions of the astrocytes and KRGE treatment (upper). Human astrocytes were subjected to oxygen–glucose deprivation (OGD) for 8 h, followed by recovery (OGD/R) with KRGE treatment for 24 h. Cell-counting kit-8 (CCK-8)-mediated cell viability is quantified (*n* = 3 independent cell cultures). (**e**) Indicated protein expressions from human astrocytes, assessed with Western blotting. β-Actin is the internal control (*n* = 5). (* *p* < 0.05, ** *p* < 0.01, and *** *p* < 0.001 between 0 μg/mL and 250 μg/mL KRGE in the OGD/R condition.) (**f**) Human astrocytes are transfected with small interfering ribonucleic acids (siRNAs) for nuclear factor erythroid 2-related factor 2 (si-Nrf2) and subjected to OGD/R with KRGE. The indicated protein expressions are detected with Western blotting (*n* = 4 independent cell cultures).

**Figure 2 cells-11-00892-f002:**
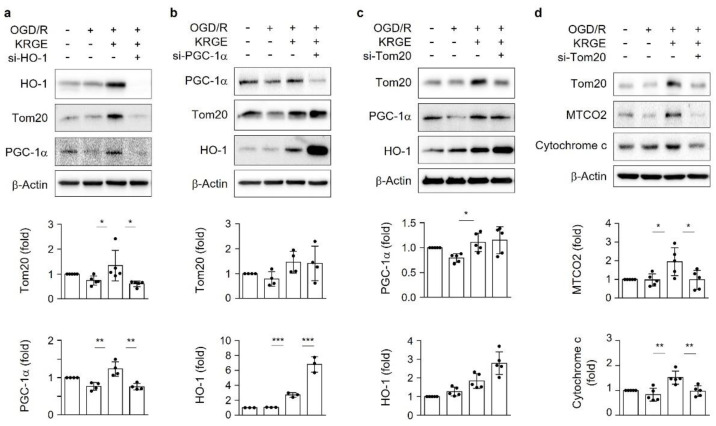
Korean red ginseng extract (KRGE) induces heme oxygenase-1 (HO-1)-mediated translocase of the outer membrane of mitochondria 20 (Tom20) in a peroxisome-proliferator-activating receptor-γ coactivator-1α (PGC-1α)-independent manner in human astrocytes. (**a**–**d**) Human astrocytes were transfected with small interfering ribonucleic acid (siRNA) for HO-1 (si-HO-1), siRNA for PGC-1α (si-PGC-1α), or siRNA for tom20 (si-Tom20) and subjected to oxygen–glucose deprivation/recovery with 250 μg/mL KRGE. We collected lysed cells from various siRNA transfections. The indicated protein levels are detected with Western blotting (*n* = 3–5 per group). * *p* < 0.05, ** *p* < 0.01, and *** *p* < 0.001.

**Figure 3 cells-11-00892-f003:**
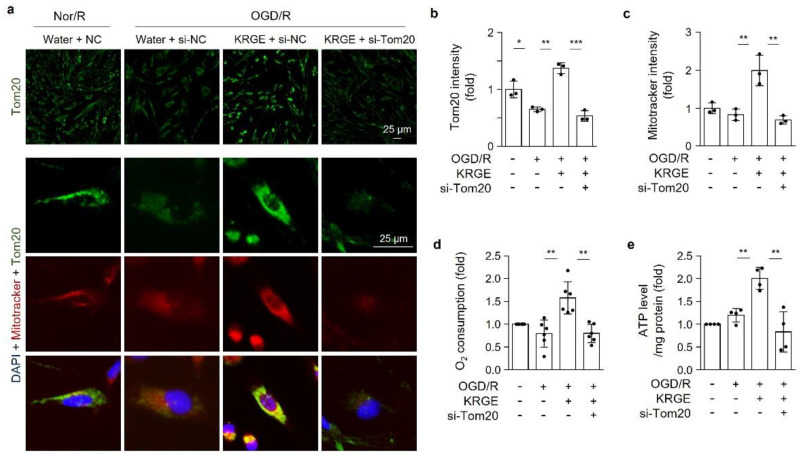
Korean red ginseng extract (KRGE) induces mitochondrial membrane potential and adenosine triphosphatase production via translocase of the outer membrane of mitochondria 20 (Tom20). (**a**–**c**) Astrocytes on 12-well plates are transfected with small interfering ribonucleic acid (siRNA) for the negative control (si-NC) or with siRNA for Tom20 (si-Tom20), followed by oxygen–glucose deprivation/recovery (OGD/R) with or without KRGE. Normoxia followed by recovery (Nor/R) is the control for OGD/R. (**a**) Representative image of Tom20 (green), Mitotracker (red; Thermo Fisher Scientific, Waltham, MA, USA), and DAPI staining (blue) in human astrocytes (*n* = 3 per group; the scale bar = 25 μm). (**b**,**c**) Relative fluorescent intensity of randomized cells, measured using ImageJ (GraphPad, San Diego, CA, USA). (**d**) Live oxygen (O_2_) consumption after probe treatment is measured over 75 min, followed by the quantification of the relative O_2_ consumption for 15 min (*n* = 6 independent experiments). (**e**) Relative ATP level/mg protein is detected and quantified (*n* = 4 independent experiments). * *p* < 0.05, ** *p* < 0.01, and *** *p* < 0.001.

**Figure 4 cells-11-00892-f004:**
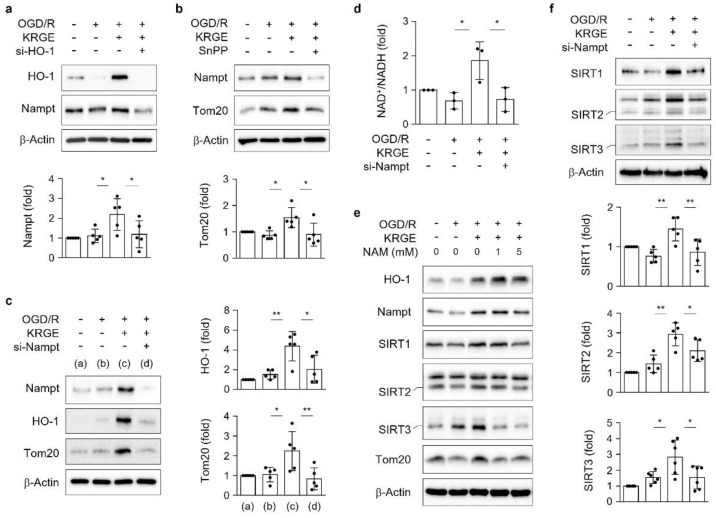
Korean red ginseng extract (KRGE) induces the heme oxygenase-1 (HO-1)-nicotinamide phosphoribosyl transferase (Nampt) circuit, thereby leading to an elevated nicotinamide adenine dinucleotide (NAD)/NAD + hydrogen (NADH) ratio in astrocytes. (**a**) Astrocytes are transfected with small interfering ribonucleic acid (siRNA) for HO-1 (si-HO-1), followed by oxygen–glucose deprivation/recovery (OGD/R) with 250 μg/mL KRGE. Protein expression is assessed with Western blotting (*n* = 5 independent experiments). (**b**) Astrocytes were incubated in OGD for 8 h, followed by 25 μM SnPP and 250 μg/mL KRGE treatment for 24 h. Indicated protein expression is obtained with Western blotting (*n* = 5 independent experiments). (**c**,**d**) Astrocytes were subjected to Nampt-specific siRNA (si-Nampt), followed by OGD/R under 250 μg/mL KRGE conditions. (**c**) Western blotting used to detect the protein expression (*n* = 5 independent experiments). (**d**) The NAD^+^/NADH ratio, as detected with an assay kit (*n* = 3 independent experiments). (**e**) Astrocytes were subjected to OGD for 8 h, followed by recovery with 1 mM and 5 mM nicotinamide (NAM) plus 250 μg/mL KRGE treatment for 24 h. Protein expression is detected with Western blotting (e.g., HO-1 (*n* = 4); Nampt (*n* = 5); silent information regulators (i.e., sirtuins [SIRTs]) SIRT1 (*n* = 5), SIRT2 (*n* = 5), and SIRT3 (*n* = 4); and translocase of the outer membrane of mitochondria 20 (Tom20) (*n* = 5)). (**f**) SIRT1, SIRT2, and SIRT3 protein expression is detected under similar conditions as that of (**a**) with Western blotting (*n* = 5 independent experiments). * *p* < 0.05 and ** *p* < 0.01.

**Figure 5 cells-11-00892-f005:**
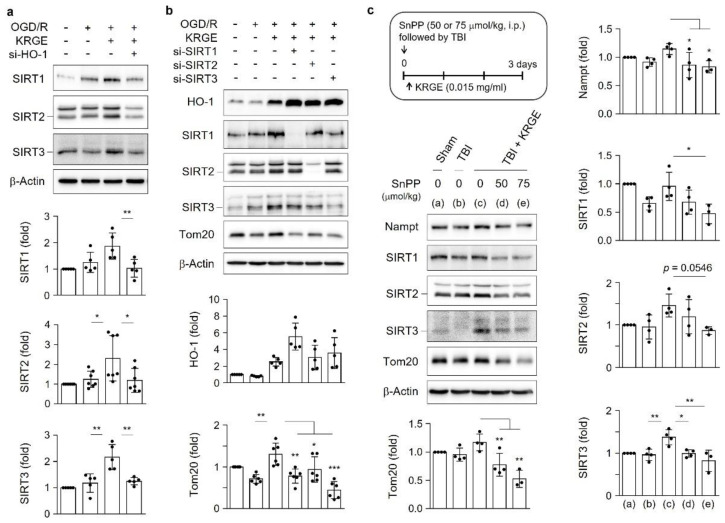
Korean red ginseng extract (KRGE) induces translocase of outer membrane of mitochondria 20 (Tom20) expression through the heme oxygenase-1 (HO-1)-class I silent information regulators (i.e., sirtuins (SIRTs)) axis. (**a**,**b**) Astrocytes are transfected with small interfering ribonucleic acid (siRNA) for HO-1 (si-HO-1), siRNA for SIRT1 (si-SIRT1), siRNA for SIRT2 (si-SIRT2), or siRNA for SIRT3 (si-SIRT3), followed by oxygen–glucose deprivation/recovery (OGD/R) with 250 μg/mL KRGE. Protein levels are assessed with Western blotting (*n* = 4–7 independent experiments). (**c**) The schematic figure presents the Sn (IV) protoporphyrin IX dichloride (SnPP), traumatic brain injury (TBI), and KRGE conditions (upper). Sham (a); TBI (b); TBI + KRGE (c); TBI + KRGE + 50 μmol/kg SnPP (d); TBI + KRGE + 75 μmol/kg SnPP (e). The indicated antibodies are detected in brain sections (approximately bregma −1 to −2) obtained from mice subjected to TBI, followed by KRGE treatment with or without SnPP, assessed with Western blotting (*n* = 3 or 4 per group). * *p* < 0.05, ** *p* < 0.01, and *** *p* < 0.001.

**Figure 6 cells-11-00892-f006:**
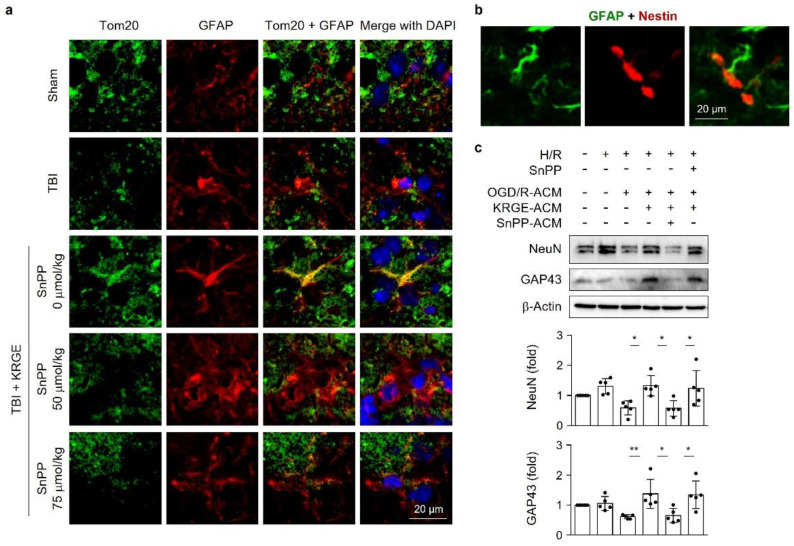
Astrocytic heme oxygenase (HO) enhances crosstalk between astrocytes and adult neural stem cells (NSCs). (**a**) Representative image of translocase of outer membrane of mitochondria 20 (Tom20, green) and glial fibrillary acidic protein (GFAP, red) in a mouse brain subjected to traumatic brain injury (TBI), followed by Korean red ginseng extract (KRGE) treatment with or without Sn(IV) protoporphyrin IX dichloride (SnPP, *n* = 3 per group. 4′,6-Diamidino-2-phenylindole (DAPI, blue) is used for nucleus detection (scale bar = 20 μm). (**b**) Representative image of GFAP (green) and Nestin (red) in a mouse brain subjected to TBI, followed by KRGE treatment (the scale bar = 20 μm). (**c**) Adult rat NSCs were incubated under hypoxia for 4 h. The cells were recovered using astrocyte-conditioned media (ACM) obtained from KRGE with or without SnPP. After 4 days of hypoxia recovery (H/R), the NSCs’ lysates underwent Western blotting. The indicated protein levels are detected in five independent experiments. * *p* < 0.05 and ** *p* < 0.01.

**Figure 7 cells-11-00892-f007:**
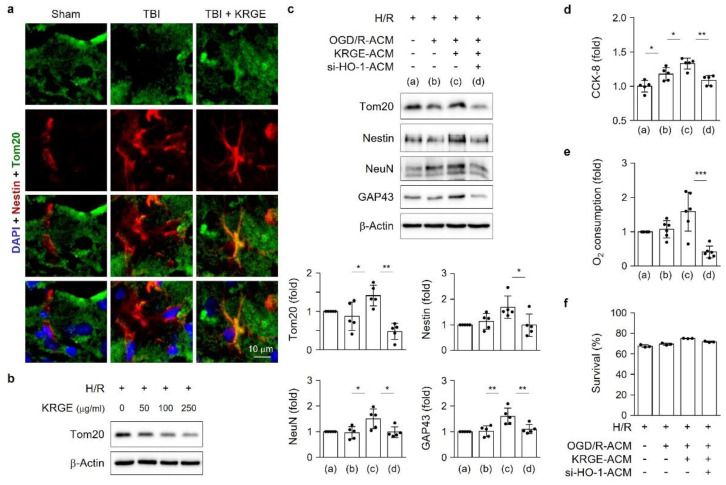
Korean red ginseng extract (KRGE) induces neural stem cell (NSC) mitochondrial functions and neuronal differentiation through astrocytic heme oxygenase-1 (HO-1). (**a**) Representative images of translocase of the outer membrane of mitochondria 20 (Tom20, green), glial fibrillary acidic protein (GFAP, red), and 4′,6-diamidino-2-phenylindole (DAPI) (blue) staining in a mouse brain subjected to traumatic brain injury (TBI), followed by KRGE treatment (*n* = 3 per group; the scale bar = 10 μm). (**b**) Direct treatment of NSCs with the indicated KRGE concentrations in Dulbecco’s modified Eagle medium (DMEM) with differentiation media were evaluated with Western blotting (*n* = 3 independent experiments). (**c**–**f**) NSCs were subjected to hypoxia for 4 h and the media were replaced with astrocyte-conditioned media (ACM) and differentiation media for 4 days after hypoxia recovery (H/R). ACM are obtained from the si-control-treated or small interfering ribonucleic acid for HO-1 (si-HO-1)-KRGE-treated cells. (**c**) NSCs assessed with Western blotting (*n* = 5 independent experiments). (**d**) Cell-counting kit 8 (CCK-8) assay (*n* = 5) and (**e**) oxygen (O_2_) consumption assay (*n* = 6) at 15 min after adding the probe. (**f**) Lactate dehydrogenase (LDH) assay (*n* = 3) is used to determine the cell survival rate (%). * *p* < 0.05, ** *p* < 0.01, and *** *p* < 0.001.

**Figure 8 cells-11-00892-f008:**
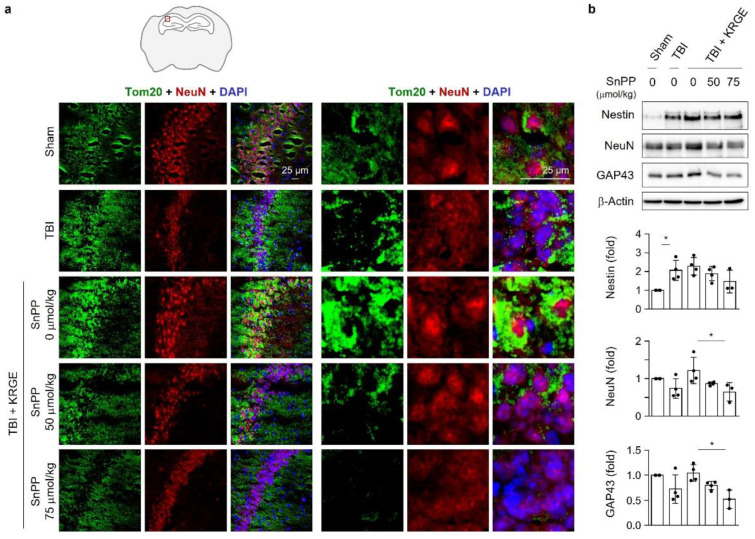
Heme oxygenase (HO) inhibition suppresses mitochondria regulating proteins. (**a**) The red box demonstrates the ipsilateral cornu ammonis 2 (CA2) region of the hippocampus after traumatic brain injury (TBI). Representative images of the translocase of outer membrane of mitochondria 20 (Tom20, green), neuronal nuclear protein (NeuN, red), and 4′,6-diamidino-2-phenylindole (DAPI) (blue) staining in a mouse brain subjected to traumatic brain injury (TBI), followed by Korean red ginseng extract (KRGE) treatment with or without Sn(IV) protoporphyrin IX dichloride (SnPP) (*n* = 3 per group; the scale bar = 25 μm). (**b**) The brain sections (approximately bregma −1 to −2) obtained from TBI, followed by KRGE treatment with or without SnPP are detected with the indicated antibodies, assessed with Western blotting (*n* = 3 or 4 per group). * *p* < 0.05.

## Data Availability

The data presented in this study are contained within the article. Original data will be made available on request from the corresponding author.

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
