# Peer review of "Dual Effects of Korean Red Ginseng on Astrocytes and Neural Stem Cells in Traumatic Brain Injury: The HO-1–Tom20 Axis as a Putative Target for Mitochondrial Function"

_cells, 2022, doi:10.3390/cells11050892_

Round 1

Reviewer 1 Report

General comment:

This manuscript, entitled “Dual effects of Korean red ginseng on astrocytes and neural stem cells in traumatic brain injury: the HO-1-Tom20 axis as a putative target for mitochondrial function.” authored by Kim et al., reports that KRGE promotes astrocytic mitochondrial functions, assessed by O2 consumption and ATP production, which could be regulated by Tom20 pathway. Astrocytic HO-1 induced mitochondrial functions; additionally, HO-1-related astrocyte-derived factor(s) may induce neuronal differentiation and mitochondrial functions of adult NSCs.Taking advantage of recent extensive uses of cell culture, immunohistochemistry, and western blotting, authors attempted to explore the KRGE-treated astrocytes, which upregulated the proliferation and neuronal differentiation of adult NSCs supposedly through astrocyte-neuronal system cooperation. They can successfully provide enough evidence through their experimental results. This kind of approach is suitable for connecting cellular biochemistry and physiology. In my opinion, this is a valuable work and suitable for publication in the Cells after addressing the following comments and questions:

Major questions:

  • Did the author consider any hypoxic effect by looking at related proteins or pathways, e.g. - hypoxia-inducible factor-1α (HIF-1α)?
  • In figure 4c, whether housekeeping control varies in concentration – how can you explain that?
  • Other than cytochrome c, whether different cytochromes are affected due to KRGE?
  • Is there any direct experimental evidence available of improving motor neuron function or saltatory motions to cope with depression or other neuro-disorder or neurological deficits?

Author Response

Reviewer #1

This manuscript, entitled “Dual effects of Korean red ginseng on astrocytes and neural stem cells in traumatic brain injury: the HO-1-Tom20 axis as a putative target for mitochondrial function.” authored by Kim et al., reports that KRGE promotes astrocytic mitochondrial functions, assessed by O2 consumption and ATP production, which could be regulated by Tom20 pathway. Astrocytic HO-1 induced mitochondrial functions; additionally, HO-1-related astrocyte-derived factor(s) may induce neuronal differentiation and mitochondrial functions of adult NSCs.Taking advantage of recent extensive uses of cell culture, immunohistochemistry, and western blotting, authors attempted to explore the KRGE-treated astrocytes, which upregulated the proliferation and neuronal differentiation of adult NSCs supposedly through astrocyte-neuronal system cooperation. They can successfully provide enough evidence through their experimental results. This kind of approach is suitable for connecting cellular biochemistry and physiology. In my opinion, this is a valuable work and suitable for publication in the Cells after addressing the following comments and questions:

Major questions: 

Comment #1. Did the author consider any hypoxic effect by looking at related proteins or pathways, e.g. - hypoxia-inducible factor-1α (HIF-1α)?

Response #1: In Figure 4b condition of manuscript, we detected hypoxia-inducible factor-1α (HIF-1α) (left: Figure 1A). We could not find any critical difference in the HIF-1α protein levels between the OGD/R and the OGD/R plus KRGE groups. In addition, we focused on the peri-injured region of the brains at 3 days after TBI. At this point, we could not detect HIF-1α expression in the KRGE-administered TBI brains (Figure 1B). Thus, we did not include HIF-1α-related pathways in this study. In our next study, we will consider investigating hypoxia-regulated signaling in core-injured regions of TBI.

Comment #2. In figure 4c, whether housekeeping control varies in concentration – how can you explain that?

Response #2: We replaced the β-actin blot data after stripping the membranes, which were detected with Tom20 in Figure 4c. We are thankful for your comment. By following your suggestion, we gained a more accurate housekeeping control.

Comment #3. Other than cytochrome c, whether different cytochromes are affected due to KRGE?

Response #3: Thank you for your comment. At this point, we did not check other cytochromes (e.g., cytochrome a, cytochrome aa3, cytochrome b5), which may be affected by KRGE. In a future study, we would like to investigate the roles of other cytochromes in KRGE-mediated energy production.

Comment #4. Is there any direct experimental evidence available of improving motor neuron function or saltatory motions to cope with depression or other neuro-disorder or neurological deficits?

Response #4: We do not have any direct experimental evidence available of KRGE improving motor neuron function or saltatory motions in TBI. We previously investigated the beneficial roles of ginseng/ginsenosides, which have been demonstrated by other researchers investigating various neurological disorders [reviewed by Kim et al. (Journal of Ginseng Research (2021) 45(5): 599-609)]. In this review, several ginsenosides (e.g., Rd, Rg1, and Rg5) resulted in behavioral improvement in mouse or rat models of neurological deficits (Table 1).

KRGE promotes adult hippocampal neurogenesis, based on the detection of BrdU/doublecortin-positive cells, and it improves learning and memory abilities, based on the findings of the Morris water maze experiment (Ryu et al., Neural Regeneration Research (2020) 15(5): 887-893). Adult neurogenesis activator combined with brain-derived neurotrophic factor (BDNF) significantly enhanced cognitive functions, which could mimic the effects of exercise on cognition, in an 5xFAD Alzheimer’s mouse model [Choi et al., Science (2018) 361(6406): eaan8821]. The treatment of human neural stem cells with BDNF and ginsenosides such as Rg1 and Rb1 similarly promoted cell survival and neurite outgrowth [Wang and Kisaalita, Journal of Neuroscience Methods (2011) 194(2): 274-282]. In addition, Rg1 is involved in BDNF-mediated neurogenesis and exhibits antidepressant activity [Jiang et al., British Journal of Pharmacology (2012) 166(6): 1872-1887].

Furthermore, KRGE pretreatment protects against acute sensorimotor deficits and promotes long-term functional recovery after ischemic stroke through the Nrf2 pathway [Liu et al., Frontiers in Cellular Neuroscience (2018) 12:74]. Taken together, KRGE or ginsenosides may improve neuronal functions. Based on these references, we revised the Introduction section, indicated in red font.

Reviewer 2 Report

Interesting topic but it is presented in a complicated way. The introduction does not introduce the problem at all, does not clarify what is already known about ginseng properties, nor does clarify to the readers why the listed analyses were carried out.

The results, in particular, the figures, are difficult to read. The problem is aggravated by several language inaccuracies. 

Concerning the methods, in particular the cytological ones, I have some doubts about tissue preparation. 20 microns is a quite thick section that in general does not allow to focus properly on preparation. In addition, tissues were dried, a procedure that usually introduces artefacts. Mitotracker in figure 3 appears as a red cloud (should appear as dots) and localise also on the nucleus (where Tom is not present).  Therefore, to consider significant the presented cytological data, at least one light micrograph of tissue/cells is necessary to prove their good condition. 

In conclusion, my opinion is that the manuscript requires an accurate revision pointing at making the content clearer and more convincing.  

Author Response

Reviewer #2

Comment #1. Interesting topic but it is presented in a complicated way. The introduction does not introduce the problem at all, does not clarify what is already known about ginseng properties, nor does clarify to the readers why the listed analyses were carried out.

Response #1: We are thankful for your suggestion. We have revised the Introduction section as per your suggestion, indicated in red font.

  1. Introduction

In central nervous system (CNS) injuries, Korean red ginseng extract (KRGE) and its components (e.g., ginsenoside) have favorable effects on neurovascular regeneration and antiinflammation [1]. Several ginsenosides have improved behavior in animal models of neurological deficits [1]. Ginsenoside Rg1 is involved in neurotrophic factor-mediated adult hippocampal neurogenesis and exhibits antidepressant activity [2]. Ginsenoside Rb1 may be protective against traumatic brain injury (TBI) by enhancing the gap junction [3]. KRGE pretreatment protects against acute sensorimotor deficits and promotes its long-term recovery after ischemic stroke through the nuclear factor erythroid 2-related factor 2 (Nrf2) pathway [4].

Neuroinflammatory brain injury, including TBI, induces the altered remodeling of astrocyte mitochondrial networks [5, 6]. Heme oxygenase-1 (HO-1), a downstream target of Nrf2, is a heme degrading enzyme that produces carbon monoxide (CO), iron, and biliverdin. In addition, biliverdin can be converted into bilirubin by biliverdin reductase [7]. HO-1 has cytoprotective effects on mitochondrial oxidative stress induced by mucosal injury in rats [8].

Nicotinamide adenine dinucleotide (NAD) salvage synthesis in mammals warrants the enzymatic activity of intracellular nicotinamide phosphoribosyl transferase (Nampt) [9]. The deletion of intracellular Nampt in the projection neurons of adult mice impairs mitochondrial function and neuromuscular junction synaptic transmission [5]. Silent information regulators (i.e., sirtuins [SIRTs]) comprise seven types (i.e., SIRT 1–7): class I [e.g., SIRT 1–3), class II (e.g., SIRT4), class III (e.g., SIRT5), and class IV (e.g., SIRT6 and SIRT7), which are NAD-dependent proteins [10-12]. SIRT1 may trigger peroxisome proliferators-activated receptor γ-coactivator-1a (PGC-1a) activation through deacetylation in skeletal muscles and provide beneficial effects on mitochondrial biogenesis [13, 14]. However, pyruvate induces mitochondria biogenesis in PGC-1a null mouse-derived myoblasts [15], which implies the existence of a molecular pathway involved in PGC-1a-independent mitochondria biogenesis.

Astrocytes in the CNS contribute to neuroprotection and energy-metabolic activity in CNS-related pathological states [5, 16]. Our previous study [6] demonstrated that KRGE induces HO-1 production in astrocytes post-TBI, which exerts crucial roles in mitochondrial activity, partly through PGC-1a. However, the roles of KRGE-induced astrocytic HO-1 in the mitochondrial components of astrocytes and neural stem cells (NSCs) have not been well studied. In this study, our novel finding was that astrocytic mitochondrial functions are considerably associated with KRGE-mediated upregulation of translocase of the outer membrane of mitochondria 20 (Tom20) through the Nrf2–HO-1 axis in which the pathway involves a PGC-1a-independent mechanism. Tom20 is transcribed in the nucleus and becomes targeted to the mitochondrial outer membrane and thereby involves the translocation of major protein precursors [17].

To investigate the unknown molecular link between HO-1 and Tom20, we evaluated NAD-dependent class I SIRTs (i.e., SIRT1, SIRT2, and SIRT3). We checked the signaling cascade among HO-1, Nampt, SIRT1, SIRT2, SIRT3, and Tom20 in KRGE-mediated mitochondrial functions in astrocytes during oxygen-glucose deprivation (OGD), followed by recovery (OGD/R). KRGE induced HO-1 expression and the consequent upregulation of Nampt-class I SIRTs–Tom20 in OGD/R-conditioned astrocytes. In an in vivo TBI model, HO inhibition by Sn(IV) protoporphyrin IX dichloride (SnPP) injection diminished the expressions of KRGE-mediated mitochondria-related proteins such as Nampt, SIRT1, SIRT2, SIRT3, and Tom20. Moreover, KRGE-mediated HO-1 induction in astrocytes also triggers intercellular communication by enhancing mitochondrial activation, neuronal differentiation, and proliferation of adult NSCs.

Comment #2. The results, in particular, the figures, are difficult to read. The problem is aggravated by several language inaccuracies.
Response #2: We have revised them and believe that they are now easier to read.

Comment #3. Concerning the methods, in particular the cytological ones, I have some doubts about tissue preparation. 20 microns is a quite thick section that in general does not allow to focus properly on preparation.

Response #3: In our previous study [Choi et al., Nature Medicine (2016) 22(11): 1335-1341], we sectioned fresh frozen brains into 20-micron thick samples (the same condition used in this study), and sectioned paraformaldehyde fixed brains into 16-micron thick samples. However, per your comment, we will attempt to reduce the thickness of the brain tissues in a future study.

Comment #4. In addition, tissues were dried, a procedure that usually introduces artefacts.
Response #4: Without acetone fixation and drying, we could not obtain clear images using anti-Tom20 antibody. We searched other protocols supplied by companies (e.g., BioLegend [San Diego, CA, USA] and Abcam [Cambridge, UK]) and tried to adjust the conditions. Based on repeated experiments, we believe this method works (i.e., acetone and 5 min evaporation at room temperature) because our results seem to be similar to those of other groups [Franco-Iborra et al., Cell Death and Disease (2018) 9:1122]. In addition, we changed the results in Figure 1a (i.e., sham) into a more accurate version of the image.

Comment #5. Mitotracker in figure 3 appears as a red cloud (should appear as dots) and localise also on the nucleus (where Tom is not present).  Therefore, to consider significant the presented cytological data, at least one light micrograph of tissue/cells is necessary to prove their good condition.
Response #5: We are thankful for your suggestion. Accordingly, we have replaced the image with more accurate Mitotracker imaging (Figure 3a). We apologize for this limitation. We surmise that clearer Mitotracker images (i.e., appearing as dots) could be obtained using a confocal microscope when cells are alive (i.e., live imaging).

In this study, however, after Mitotracker treatment, the cells were fixed and stained with anti-Tom20 antibody. Images were then obtained by using an inverted phase contrast microscope (Eclipse Ti2-U; Nikon, Tokyo, Japan). The intensity of Mitotracker in the OGD/R plus KRGE group is the strongest, compared to the intensity of the other groups.

To demonstrate the cells’ condition, we added 4′,6-diamidino-2-phenylindole (DAPI) staining images in Supplementary Figure A2. The density and shape of nuclei appears to be similar in all groups.

Supplementary Figure A2. Diamidino-2-phenylindole (DAPI) staining findings (in blue).

Comment #6. In conclusion, my opinion is that the manuscript requires an accurate revision pointing at making the content clearer and more convincing.  

Response #6: We have attempted to make the content clearer and more convincing by replacing images and adding explanations. Thank you for your critical comments.

Round 2

Reviewer 2 Report

The manuscript is significantly improved and is now worth publication.